# Dynamic Mode Decomposition with Reproducing Kernels for Koopman Spectral Analysis

**Yoshinobu Kawahara**[ab]
[a] The Institute of Scientific and Industrial Research, Osaka University
[b] Center for Advanced Integrated Intelligence Research, RIKEN
ykawahara@sanken.osaka-u.ac.jp

## Abstract

A spectral analysis of the Koopman operator, which is an infinite dimensional linear operator on an observable, gives a (modal) description of the global behavior of a nonlinear dynamical system without any explicit prior knowledge of its governing equations. In this paper, we consider a spectral analysis of the Koopman operator in a reproducing kernel Hilbert space (RKHS). We propose a modal decomposition algorithm to perform the analysis using finite-length data sequences generated from a nonlinear system. The algorithm is in essence reduced to the calculation of a set of orthogonal bases for the Krylov matrix in RKHS and the eigendecomposition of the projection of the Koopman operator onto the subspace spanned by the bases. The algorithm returns a decomposition of the dynamics into a finite number of modes, and thus it can be thought of as a feature extraction procedure for a nonlinear dynamical system. Therefore, we further consider applications in machine learning using extracted features with the presented analysis. We illustrate the method on the applications using synthetic and real-world data.

## 1 Introduction

Modeling nonlinear dynamical systems using data is fundamental in a variety of engineering and scientific fields. In machine learning, the problem of learning dynamical systems has been actively discussed, and several Bayesian approaches have been proposed [11, 34]. In the fields of physics, one popular approach for this purpose is the decomposition methods that factorize the dynamics into modes based on some criterion from the data. For example, proper orthogonal decomposition (POD) (see, for example, [12]), which generates orthogonal modes that optimally capture the vector energy of a given dataset, has been extensively applied to complex phenomena in physics [5, 22] even though this method is currently known to have several drawbacks. The so-called spectral method for dynamical systems [15, 31, 17], which is often discussed in machine learning, is closely related to this type of technique, where one aims to estimate a prediction model rather than understand the dynamics by examining the obtained modes.

Among the decomposition techniques, *dynamic mode decomposition (DMD)* [25, 26] has recently attracted attention in the field of physics, such as flow mechanics, and in engineering, and has been applied to data obtained from complex phenomena [2, 4, 6, 10, 21, 25, 27, 32]. DMD approximates the spectra of the Koopman operator [16], which is an infinite-dimensional linear operator that represents nonlinear and finite-dimensional dynamics without linearization. While POD just finds the principal directions in a dataset, DMD can yield direct information concerning the dynamics such as growth rates and the frequencies of the dynamics.

In this paper, we consider a spectral analysis of the Koopman operator in reproducing kernel Hilbert spaces (RKHSs) for a nonlinear dynamical system

$$\boldsymbol{x}_{t+1} = \boldsymbol{f}(\boldsymbol{x}_t), \tag{1}$$

where $\boldsymbol{x} \in \mathcal{M}$ is the state vector on a finite-dimensional manifold $\mathcal{M} \subseteq \mathbb{R}^d$, and $\boldsymbol{f}$ is a (possibly, nonlinear) state-transition function. We present a modal decomposition algorithm to perform this,

which is in principle reduced to the calculation of a set of orthogonal bases for the Krylov matrix in RKHS and the eigendecomposition of the projection of the Koopman operator onto the subspace spanned by the bases. Although existing DMD algorithms can conceptually be thought of as producing an approximation of the eigenfunctions of the Koopman operator using a set of linear monomials of observables (or the pre-determined functional maps of observables) as basis functions, which is analogous to a one-term Taylor expansion at each point, our algorithm gives an approximation with a set of nonlinear basis functions due to the expressiveness of kernel functions. The proposed algorithm provides a modal decomposition of the dynamics into a finite number of modes, and thus it could be considered as a feature extraction procedure for a nonlinear dynamical system. Therefore, we consider applications using extracted features from our analysis such as state prediction, sequential change-point detection, and dynamics recognition. We illustrate our method on the applications using synthetic and real-world data.

The remainder of this paper is organized as follows. In Section 2, we briefly review the spectral analysis of nonlinear dynamical systems with the Koopman operator and DMD. In Section 3, we extend the analysis with reproducing kernels, and provide a modal decomposition algorithm to perform this analysis based on the equivalent principle of DMD. Although this method is mathematically correct, a practical implementation could yield an ill-conditioned algorithm. Therefore, in Section 4, we describe a way to robustly it by projecting data onto the POD directions. In Section 5, we describe related works. In Section 6, we show some empirical examples by the proposed algorithm and, in Section 7, we describe several applications using extracted features with empirical results. Finally, we conclude the paper in Section 8.

## 2   The Koopman Operator and Dynamic Mode Decomposition

Consider a discrete-time nonlinear dynamical system (1). *The Koopman operator* [16], which we denote here by $\mathcal{K}$, is an *infinite-dimensional* linear operator that acts on a scalar function $g_i \colon \mathcal{M} \to \mathbb{C}$, mapping $g_i$ to a new function $\mathcal{K} g_i$ given as follows:

$$(\mathcal{K} g_i)(\boldsymbol{x}) = g_i \circ \boldsymbol{f}(\boldsymbol{x}), \tag{2}$$

where $\circ$ denotes the composition of $g_i$ with $\boldsymbol{f}$. We see that $\mathcal{K}$ acts linearly on the function $g_i$, even though the dynamics defined by $\boldsymbol{f}$ may be nonlinear. Since $\mathcal{K}$ is a linear operator, it has, in general, an eigendecomposition

$$\mathcal{K} \varphi_j(\boldsymbol{x}) = \lambda_j \varphi_j(\boldsymbol{x}), \tag{3}$$

where $\lambda_j \in \mathbb{C}$ is the $j$-th eigenvalue (called the *Koopman eigenvalue*) and $\varphi_j$ is the corresponding eigenfunction (called the *Koopman eigenfunction*). We denote the concatenation of $g_i$ as $\boldsymbol{g} := [g_1, \ldots, g_p]^\top$. If each $g_i$ lies within the span of the eigenfunctions $\varphi_j$, we can expand the vector-valued $\boldsymbol{g}$ in terms of these eigenfunctions as

$$\boldsymbol{g}(\boldsymbol{x}) = \sum_{j=1}^{\infty} \varphi_j(\boldsymbol{x}) \boldsymbol{u}_j, \tag{4}$$

where $\boldsymbol{u}_j$ is a set of vector coefficients called *Koopman modes*. Then, by the iterative applications of Eqs. (2) and (3), we obtain

$$\boldsymbol{g} \circ \boldsymbol{f}^l(\boldsymbol{x}) = \sum_{j=1}^{\infty} \lambda_j^l \varphi_j(\boldsymbol{x}) \boldsymbol{u}_j, \tag{5}$$

where $\boldsymbol{f}^l$ is the $l$-time compositions of $\boldsymbol{f}$. Therefore, $\lambda_j$ characterizes the temporal behavior of the corresponding Koopman mode $\boldsymbol{u}_j$, i.e., the phase of $\lambda_j$ determines its frequency, and the magnitude determines the growth rate of the dynamics. Note that, for a system evolving on an attractor, the Koopman eigenvalues always lie on a unit circle [20].

DMD [25, 26] (and its variants) is a popular approach for estimating the approximations of $\lambda_j$ and $\boldsymbol{u}_j$ from a finite-length data sequence $\boldsymbol{y}_0, \boldsymbol{y}_1, \ldots, \boldsymbol{y}_\tau (\in \mathbb{R}^p)$, where we denote $\boldsymbol{y}_t := \boldsymbol{g}(\boldsymbol{x}_t)$. DMD can fundamentally be considered as a special use of the Arnoldi method [1]. That is, using the empirical Ritz values $\tilde{\lambda}_j$ and vectors $\boldsymbol{v}_j$ obtained by the Arnoldi method when regarding the subspace spanned by $\boldsymbol{y}_0, \ldots, \boldsymbol{y}_{\tau-1}$ as the Krylov subspace for $\boldsymbol{y}_0$ (and implicitly for some matrix $A \in \mathbb{R}^{p \times p}$), it is shown that the observables are expressed as

$$\boldsymbol{y}_t = \sum_{j=1}^{\tau} \tilde{\lambda}_j^t \boldsymbol{v}_j \ \ (t = 0, \ldots, \tau - 1), \ \text{ and} \tag{6a}$$

$$\boldsymbol{y}_\tau = \sum_{j=1}^{\tau} \tilde{\lambda}_j^\tau \boldsymbol{v}_j + \boldsymbol{r} \ \text{ where } \ \boldsymbol{r} \perp \mathrm{span}\{\boldsymbol{y}_0, \ldots, \boldsymbol{y}_{\tau-1}\}. \tag{6b}$$

Comparing Eq. (6a) with Eq. (5) infers that the empirical Ritz values $\tilde{\lambda}_j$ and vectors $\boldsymbol{v}_j$ behave in precisely the same manner as the Koopman eigenvalues $\lambda_j$ and modes $\boldsymbol{u}_j$ ($\varphi_j(\boldsymbol{x}_0) \boldsymbol{u}_j$), but for the

finite sum in Eq. (6a) instead of the infinite sum in Eq. (5). Note that, for $r = 0$ in Eq. (6b) (which could happen when the data are sufficiently large), the approximate modes are indistinguishable from the true Koopman eigenvalues and modes (as far as the data points are concerned), with the expansion (5) comprising only a finite number of terms.

## 3 Dynamic Mode Decomposition with Reproducing Kernels

As described above, the estimation of the Koopman mode by DMD (and its variants) can capture the nonlinear dynamics from finite-length data sequences generated from a dynamical system. Conceptually, DMD can be considered as producing an approximation of the Koopman eigenfunctions using a set of linear monomials of observables as basis functions, which is analogous to a one-term Taylor expansion at each point. In situations where eigenfunctions can be accurately approximated using linear monomials (e.g., in a small neighborhood of a stable fixed point), DMD will produce an accurate local approximation of the Koopman eigenfunctions. However, this is certainly not applicable to all systems (in particular, beyond the region of validity for local linearization). Here, we extend the Koopman spectral analysis with reproducing kernels to approximate the Koopman eigenfunctions with *richer* basis functions. We provide a modal decomposition algorithm to perform this analysis based on the equivalent principle with DMD.

Let $\mathcal{H}$ be the RKHS embedded with the dot product $\langle \cdot, \cdot \rangle_{\mathcal{H}}$ (we abbreviate $\langle \cdot, \cdot \rangle_{\mathcal{H}}$ as $\langle \cdot, \cdot \rangle$ for simplicity) and a positive definite kernel $k$. Additionally, let $\phi \colon \mathcal{M} \to \mathcal{H}$. Then, we define the Koopman operator on the feature map $\phi$ by

$$(\mathcal{K}_{\mathcal{H}}\phi)(\boldsymbol{x}) = \phi \circ \boldsymbol{f}(\boldsymbol{x}). \tag{7}$$

Thus, the Koopman operator $\mathcal{K}_{\mathcal{H}}$ is a linear operator in $\mathcal{H}$. Note that almost of the theoretical claims in this and the next sections do not necessarily require $\phi$ to be in RKHS (it is sufficient that $\phi$ stays in a Hilbert space). However, this assumption should perform the calculation in practice (as described in the last parts of this and the next sections). Therefore, we proceed with this assumption in the following parts. We denote by $\varphi_j$ the $j$-th eigenfunction of $\mathcal{K}_{\mathcal{H}}$ with the corresponding eigenvalue $\lambda_j$. Also, we define $\Phi := \mathrm{span}\{\phi(\boldsymbol{x}) \colon \boldsymbol{x} \in \mathcal{M}\}$.

We first expand the notions, such as the Ritz values and vectors, that appear in DMD with reproducing kernels. Suppose we have a sequence $\boldsymbol{x}_0, \boldsymbol{x}_1, \dots, \boldsymbol{x}_{\tau}$. The Krylov subspace for $\phi(\boldsymbol{x}_0)$ is defined as the subspace spanned by $\phi(\boldsymbol{x}_0), (\mathcal{K}_{\mathcal{H}}\phi)(\boldsymbol{x}_0), \dots, (\mathcal{K}_{\mathcal{H}}^{\tau-1}\phi)(\boldsymbol{x}_0)$. Note that this is identical to the one spanned by $\phi(\boldsymbol{x}_0), \dots, \phi(\boldsymbol{x}_{\tau-1})$, whose corresponding Krylov matrix is given by

$$\mathcal{M}_{\tau} = [\phi(\boldsymbol{x}_0) \ \cdots \ \phi(\boldsymbol{x}_{\tau-1})]. \tag{8}$$

Therefore, if we denote a set of $\tau$ orthogonal bases of the Krylov subspace by $q_1, \dots, q_{\tau} \ (\in \mathcal{H})$ (obtained from the Gram-Schmidt orthogonalization described below), then the orthogonal projection of $\mathcal{K}_{\mathcal{H}}$ onto $\mathcal{M}_{\tau}$ is given by $\mathcal{P}_{\tau} = \mathcal{Q}_{\tau}^* \mathcal{K}_{\mathcal{H}} \mathcal{Q}_{\tau}$, where $\mathcal{Q}_{\tau} = [q_1 \ \cdots \ q_{\tau}]$ and $\mathcal{Q}_{\tau}^*$ indicates the Hermitian transpose of $\mathcal{Q}_{\tau}$. Consequently, the empirical Ritz values and vectors are defined as the eigenvalues and vectors of $\mathcal{P}_{\tau}$, respectively. Now, we have the following theorem:

**Theorem 1.** *Consider a sequence $\phi(\boldsymbol{x}_0), \phi(\boldsymbol{x}_1), \dots, \phi(\boldsymbol{x}_{\tau})$, and let $\tilde{\lambda}_j$ and $\tilde{\varphi}_j$ be the empirical Ritz values and vectors for this sequence. Assume that $\tilde{\lambda}_j$'s are distinct. Then, we have*

$$\phi(\boldsymbol{x}_t) = \sum_{j=1}^{\tau} \tilde{\lambda}_j^t \tilde{\varphi}_j \ \ (t = 0, \dots, \tau - 1), \ \ and \tag{9a}$$

$$\phi(\boldsymbol{x}_{\tau}) = \sum_{j=1}^{\tau} \tilde{\lambda}_j^{\tau} \tilde{\varphi}_j + \psi \ \ where \ \ \psi \perp \mathrm{span}\{\phi(\boldsymbol{x}_0), \dots, \phi(\boldsymbol{x}_{\tau-1})\}. \tag{9b}$$

*Proof.* Let $\mathcal{M}_{\tau} = \mathcal{Q}_{\tau} R \ (R \in \mathbb{C}^{\tau \times \tau})$ be the Gram-Schmidt QR decomposition of $\mathcal{M}_{\tau}$. Then, the companion matrix (rational canonical form) of $\mathcal{P}_{\tau}$ is given as $F := R^{-1} \mathcal{P}_{\tau} R$. Note that the sets of eigenvalues of $\mathcal{P}_{\tau}$ and $F$ are equivalent. Since $F$ is a companion matrix and $\tilde{\lambda}_j$'s are distinct, $F$ can be diagonalized in the form $F = T^{-1} \tilde{\Lambda} T$, where $\tilde{\Lambda}$ is a diagonal matrix with $\tilde{\lambda}_1, \dots, \tilde{\lambda}_{\tau}$ and $T$ is a Vandermonde matrix defined by $T_{ij} = \tilde{\lambda}_i^{j-1}$. Therefore, the empirical Ritz vectors $\tilde{\varphi}_j$ are obtained as the columns of $V = \mathcal{M}_{\tau} T^{-1}$. This proves Eq. (9a). Suppose a linear expansion of $\phi(\boldsymbol{x}_{\tau})$ is represented as

$$\phi(\boldsymbol{x}_{\tau}) = \mathcal{M}_{\tau} \boldsymbol{c} + \psi \ \ where \ \ \psi \perp \mathrm{span}\{\phi(\boldsymbol{x}_0), \dots, \phi(\boldsymbol{x}_{\tau-1})\}. \tag{10}$$

Since $F = R^{-1} \mathcal{P}_{\tau} R = \mathcal{M}_{\tau}^{-1} \mathcal{K}_{\mathcal{H}} \mathcal{M}_{\tau}$ (therefore, $\mathcal{M}_{\tau} F = \mathcal{K}_{\mathcal{H}} \mathcal{M}_{\tau}$), the first term is given by the last column of $\mathcal{M}_{\tau} F = \mathcal{M}_{\tau} T^{-1} \tilde{\Lambda} T = V \tilde{\Lambda} T$. This proves Eq. (9b). $\qquad \square$

This theorem gives an extension of DMD via the Gram-Schmidt QR decomposition in the feature space. Although in Step (2), the Gram-Schmidt QR orthogonalization is performed in RKHS, this calculation can be reduced to operations on a Gram matrix due to the reproducing property of kernel functions.

(1) Define $\mathcal{M}_\tau$ by Eq. (8) and $\mathcal{M}_+ := [\phi(\boldsymbol{x}_1), \ldots, \phi(\boldsymbol{x}_\tau)]$.

(2) Calculate the Gram-Schmidt QR decomposition $\mathcal{M}_\tau = \mathcal{Q}_\tau R$ (e.g., refer to Section 5.2 of [29]).

(3) Calculate the eigendecomposition of $R^{-1}\mathcal{Q}_\tau^*\mathcal{M}_+(=F) = T^{-1}\tilde{\Lambda}T$, where each diagonal element of $\tilde{\Lambda}$ gives $\tilde{\lambda}_j$.

(4) Define $\tilde{\varphi}_j$ to be the columns of $\mathcal{M}_\tau T^{-1}$.

The original DMD algorithm (and its variants) produce an approximation of the eigenfunctions of the Koopman operator in Eq. (2) using the set of linear monomials of observables as basis functions. In contrast, because the above algorithm works with operations directly in the functional space, the Koopman operator defined in Eq. (7) is identical to the transition operator on an observable. Therefore, the eigenfunctions of the Koopman operator are fully recovered if the Krylov subspace is sufficiently large, i.e., $\phi(\boldsymbol{x}_\tau)$ is also in $\mathrm{span}\{\phi(\boldsymbol{x}_0), \ldots, \phi(\boldsymbol{x}_{\tau-1})\}$ (or $\psi = 0$).

## 4    Robustifying with POD Bases

Although the above decomposition based on the Gram-Schmidt orthogonalization is mathematically correct, a practical implementation could yield an ill-conditioned algorithm that is often incapable of extracting multiple modes. A similar issue has been well known for DMD [26], where one needs to adopt a way to robustify DMD by projecting data onto the (truncated) POD directions [8, 33]. Here, we discuss a similar modification of our principle with the POD basis.

First, consider kernel PCA [28] on $\boldsymbol{x}_0, \boldsymbol{x}_1, \ldots, \boldsymbol{x}_{\tau-1}$: Let $\bar{G} = BSB^*$ be the eigen-decomposition of the centered Gram matrix $\bar{G} = \mathbf{H}G\mathbf{H} = G - \mathbf{1}_\tau G - G\mathbf{1}_\tau + \mathbf{1}_\tau G\mathbf{1}_\tau$, where $G = \mathcal{M}_\tau^*\mathcal{M}_\tau$ is the Gram matrix for the data, $\mathbf{H} = \mathbf{I} - \mathbf{1}_\tau$ and $\mathbf{1}_\tau$ is a $\tau$-by-$\tau$ matrix for which each element takes the value $1/\tau$. Suppose the eigenvalues and eigenvectors can be truncated accordingly based on the magnitudes of the eigenvalues, which results in $\bar{G} \approx \bar{B}\bar{S}\bar{B}^*$ where $p$ $(\leq \tau)$ eigenvalues are adopted. Denote the $j$-th column of $\bar{B}$ by $\boldsymbol{\beta}_j$ and let $\bar{\phi}(\boldsymbol{x}_i) = \phi(\boldsymbol{x}_i) - \phi_c$, where $\phi_c = \sum_{j=0}^{\tau-1} \phi(\boldsymbol{x}_j)$. A principal orthogonal direction in the feature space is then given by $\nu_j = \sum_{i=0}^{\tau-1} \alpha_{j,i}\bar{\phi}(\boldsymbol{x}_i) = \mathcal{M}_\tau\mathbf{H}\boldsymbol{\alpha}_j$ ($j = 1, \ldots, p$), where $\boldsymbol{\alpha}_j = \bar{S}_{jj}^{-1/2}\boldsymbol{\beta}_j$. Let $\mathcal{U} = [\nu_1, \ldots, \nu_p]$ $(= \mathcal{M}_\tau\mathbf{H}\bar{B}\bar{S}^{-1/2})$. Since $\mathcal{M}_+ = \mathcal{K}_\mathcal{H}\mathcal{M}_\tau$, the projection of $\mathcal{K}_\mathcal{H}$ onto the space spanned by $\nu_j$ is given as

$$\hat{F} := \mathcal{U}^*\mathcal{K}_\mathcal{H}\mathcal{U} = \bar{S}^{-1/2}\bar{B}^*\mathbf{H}(\mathcal{M}_\tau^*\mathcal{M}_+)\mathbf{H}\bar{B}\bar{S}^{-1/2}. \tag{11}$$

Note that the $(i,j)$-the element of the matrix $(\mathcal{M}_\tau^*\mathcal{M}_+)$ is given by $k(\boldsymbol{x}_{i-1}, \boldsymbol{x}_j)$. Then, if we let $\hat{F} = \hat{T}^{-1}\hat{\Lambda}\hat{T}$ be the eigendecomposition of $\hat{F}$, then

$$\bar{\varphi}_j = \mathcal{U}\boldsymbol{b}_j = \mathcal{M}_\tau\mathbf{H}\bar{B}\bar{S}^{-1/2}\boldsymbol{b}_j,$$

where $\boldsymbol{b}_j$ is the $j$-th column of $\hat{T}^{-1}$, can be used as an alternative to the empirical Ritz vector $\tilde{\varphi}_j$. That is, we have the following theorem:

**Theorem 2.** *Assume that $\varphi_j \in \Phi$, so that $\varphi_j(\boldsymbol{x}) = \langle \phi(\boldsymbol{x}), \kappa_j \rangle$ for some $\kappa_j \in \mathcal{H}$ and $\forall \boldsymbol{x} \in \mathcal{M}$. If $\kappa_j$ is in the subspace spanned by the columns of $\mathcal{U}$, so that $\kappa_j = \mathcal{U}\boldsymbol{a}_j$ for some $\boldsymbol{a}_j \in \mathbb{C}^p$, then $\boldsymbol{a}_j$ is a left eigenvector of $\hat{F}$ with eigenvalue $\lambda_j$, and also we have*

$$\phi(\boldsymbol{x}) = \sum_{j=1}^p \varphi_j(\boldsymbol{x})\bar{\varphi}_j. \tag{12}$$

*Proof.* Since $\mathcal{K}_\mathcal{H}\varphi_j = \lambda_j\varphi_j$, we have $\langle \phi(\boldsymbol{f}(\boldsymbol{x})), \kappa_j \rangle = \lambda_j \langle \phi(\boldsymbol{x}), \kappa_j \rangle$. Thus, from the assumption,

$$\langle \phi(\boldsymbol{f}(\boldsymbol{x})), \mathcal{U}\boldsymbol{a}_j \rangle = \lambda_j \langle \phi(\boldsymbol{x}), \mathcal{U}\boldsymbol{a}_j \rangle.$$

By evaluating at $\boldsymbol{x}_0, \boldsymbol{x}_1, \ldots, \boldsymbol{x}_{\tau-1}$ and then stacking into matrices, we have

$$(\mathcal{U}\boldsymbol{a}_j)^*\mathcal{M}_+ = \lambda_j(\mathcal{U}\boldsymbol{a}_j)^*\mathcal{M}_\tau.$$

If we multiply $\mathbf{H}\bar{G}^{-1}\mathbf{H}\mathcal{M}_\tau^*\mathcal{U}$ from the righthand side, this gives

$$\boldsymbol{a}_j^*\mathcal{U}^*\mathcal{M}_+\mathbf{H}\bar{G}^{-1}\mathbf{H}\mathcal{M}_\tau^*\mathcal{U} = \lambda_j\boldsymbol{a}_j^*\mathcal{U}^*\mathcal{M}_\tau\mathbf{H}\bar{G}^{-1}\mathbf{H}\mathcal{M}_\tau^*\mathcal{U} = \lambda_j\boldsymbol{a}_j^*.$$

Since $\mathcal{U}^* \mathcal{M}_+ \mathbf{H} \bar{G}^{-1} \mathbf{H} \mathcal{M}_\tau^* \mathcal{U} = \mathcal{U}^* \mathcal{K}_\mathcal{H} \mathcal{U}(= \hat{F})$, this means $\boldsymbol{a}_j$ is a left eigenvector of $\hat{F}$ with eigenvalue $\lambda_j$. Let $\boldsymbol{b}_j$ be a (right) eigenvector of $\hat{F}$ with eigenvalue $\lambda_j$ and the corresponding left eigenvector $\boldsymbol{a}_j$. Assuming these have been normalized so that $\boldsymbol{a}_j^* \boldsymbol{b}_j = \delta_{ij}$, then any vector $\boldsymbol{h} \in \mathbb{C}^p$ can be written as $\boldsymbol{h} = \sum_{j=1}^p (\boldsymbol{a}_j^* \boldsymbol{h}) \boldsymbol{b}_j$. Applying this to $\mathcal{U}^* \phi(\boldsymbol{x})$ gives

$$\mathcal{U}^* \phi(\boldsymbol{x}) = \sum_{j=1}^p (\boldsymbol{a}_j^* \mathcal{U}^* \phi(\boldsymbol{x})) \boldsymbol{b}_j. = \sum_{j=1}^p \varphi_j(\boldsymbol{x}) \boldsymbol{b}_j$$

Since $\boldsymbol{b}_j = (\mathcal{U}^* \mathcal{U}) \boldsymbol{b}_j = \mathcal{U}^* \bar{\varphi}_j$, this proves Eq. (12). $\qquad\square$

This theorem clearly gives the connection between the eigenvalues/eigenvectors found by the above procedure and the Koopman eigenvalues/eigenfunctions. The assumptions in the theorem means that the data are sufficiently rich and thus a set of the kernel principal components gives a good approximation of the representation with the Koopman eigenfunctions. As in the case of Eq. (5), by the iterative applications of Eq. (3), we obtain

$$\phi(\boldsymbol{x}_t) = \sum_{j=1}^p \lambda_j^t \varphi_j(\boldsymbol{x}_0) \bar{\varphi}_j. \tag{13}$$

The procedure for the robustified variant of the DMD is summarized as follows.[1]

(1) Define $\mathcal{M}_\tau$ and calculate the centered Gram matrix $\bar{G} = \mathbf{H} \mathcal{M}_\tau^* \mathcal{M}_\tau \mathbf{H}$.

(2) Calculate the eigendecomposition $\bar{G} \approx \bar{B} \bar{S} \bar{B}^*$, which gives the kernel principal directions $\mathcal{U}$.

(3) Calculate $\hat{F}$ as in Eq. (11) and its eigendecomposition $\hat{F} = \hat{T}^{-1} \hat{\Lambda} \hat{T}$, where each diagonal element of $\hat{\Lambda}$ gives $\lambda_j$.

(4) Define $\bar{\varphi}_j$ to be the columns of $\mathcal{M}_\tau \mathbf{H} \bar{B} \bar{S}^{-1/2} \hat{T}^{-1}$.

Unlike the procedure described in Section 3, the above procedure can perform the truncation of eigenvectors corresponding to small singular values. As well as DMD, this step becomes beneficial in practice when the Gram matrix $G$, in our case, is rank-deficient or nearly so.

*Remark*: Although we assumed that data is a consecutive sequence for demonstrating the correctness of the algorithm, as evident from the above steps, the estimation procedure itself does not necessarily require a sequence but rather a collection of pairs of consecutive observables $\{(\boldsymbol{x}_1^{(i)}, \boldsymbol{x}_2^{(i)})\}_{i=1}^\tau$, where each pair is supposed to be $\boldsymbol{x}_2^{(i)} = \boldsymbol{f}(\boldsymbol{x}_1^{(i)})$, with the appropriate definitions of $\mathcal{M}_\tau$ and $\mathcal{M}_+$.

## 5 Related Works

Spectral analysis (or, referred as the decomposition technique) for dynamical systems is a popular approach aimed at extracting information concerning (low-dimensional) dynamics from data. Common techniques include global eigenmodes for linearized dynamics (see, e.g., [3]), discrete Fourier transforms, POD for nonlinear dynamics [30, 12], and balancing modes for linear systems [24] as well as multiple variants of these techniques, such as those using shift modes [22] in conjunction with POD modes. In particular, POD, which is in principle equivalent to principal component analysis, has been extensively applied to the analysis of physical phenomena [5, 22] even though it suffers from numerous known issues, including the possibility of principal directions in a set of data may not necessarily correspond to the dynamically important ones.

DMD has recently attracted considerable attention in physics such as fluid mechanics [2, 10, 21, 25, 27] and in engineering fields [4, 6, 32]. Unlike POD (and its variants), DMD yields direct information about the dynamics such as growth rates and frequencies associated with each mode, which can be obtained from the magnitude and phase of each corresponding eigenvalue of the Koopman operator. However, the original DMD has several numerical disadvantages related to the accuracy of the approximate expressions of the Koopman eigenfunctions from data. Therefore, several variants of DMD have been proposed to rectify this point, including exact DMD [33] and optimized DMD [8]. Jovanović et al. proposed sparsity-promoting DMD [13], which provides a framework for the approximation of the Koopman eigenfunctions with fewer bases. Williams et al. proposed extended DMD [35], which works on pre-determined basis functions instead of the monomials of observables. Although in extended DMD the Koopman mode is defined as the eigenvector of the corresponding operator of coefficients on basis functions, the resulting procedure is similar to the robust-version of our algorithm.

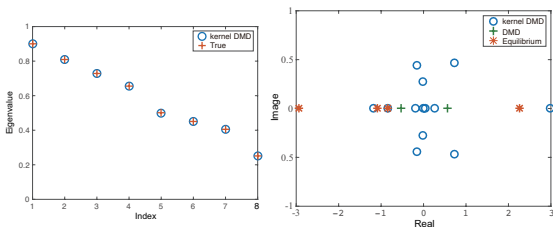

Figure 1: Estimated eigenvalues with the data from the toy system (left) and the Hénon map (right).

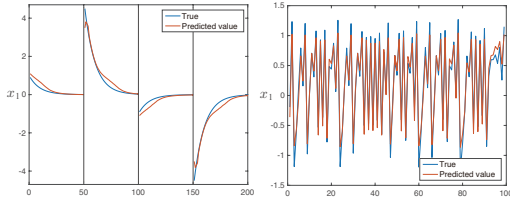

Figure 2: Examples of the true versus (1-step) predicted values via the proposed method for the toy system (left) and the Hénon map (right).

In system control, *subspace identification* [23, 14], or called the eigensystem realization method, has been a popular approach to modeling of dynamical systems. This method basically identifies low-dimensional (hidden) states as canonical vectors determined by canonical correlation analysis, and estimates parameters in the governing system using the state estimates. This type of method is known as a *spectral method* for dynamical systems in the machine learning community and has recently been applied to several types of systems such as variants of hidden Markov models [31, 19], nonlinear dynamical systems [15], and predictive state-representation [17]. The relation between DMD and other methods, particularly the eigensystem realization method, is an interesting open problem. This is briefly mentioned in [33] but it would require further investigation in future studies.

## 6 Empirical Example

To illustrate how our algorithm works, we here consider two examples: a toy nonlinear system given by $x_{t+1} = 0.9x_t$, $y_{t+1} = 0.5y_t + (0.9^2 - 0.5)x_t^2$, and one of the well-known chaotic maps, called the Hénon map ($x_{t+1} = 1 - ax_t^2 + y_t$, $y_{t+1} = bx_t$), which was originally presented by Hénon as a simplified model of the Poincaré section of the Lorenz attractor. As for the toy one, the two eigenvalues are 0.5 and 0.9 with the corresponding eigenfunctions $\varphi_{0.9} = x_t$ and $\varphi_{0.5} = y_t - x_t^2$, respectively. And as for the Hénon map, we set the parameters as $a = 1.4$, $b = 0.3$. It is known that this map has two equilibrium points $(-1.13135, -0.339406)$ and $(0.631354, 0.189406)$, whose corresponding eigenvalues are $2.25982$ and $-1.09203$, and $-2.92374$ and $-0.844054$.

We generated samples according to these systems with several initial conditions and then applied the presented procedure to estimate the Koopman modes. We used the polynomial kernel of degree three for the toy system, and the Gaussian kernel with width 1 for the Hénon map, respectively. The graphs in Fig. 1 show the estimated eigenvalues for two cases. As seen from the left graph, the eigenvalues for the toy system were precisely estimated. Meanwhile, from the right graph, the part of the eigenvalues of the equilibrium points seem to be approximately estimated by the algorithm.

## 7 Applications

The above algorithm provides a decomposition of the dynamics into a finite number of modes, and therefore, could be considered as a feature extraction procedure for a nonlinear dynamical system. This would be useful to directly understand dominant characteristics of the dynamics, as done in scientific fields with DMD [2, 10, 21, 25, 27]. However, here we consider some examples of applications using extracted features with the proposed analysis; prediction, sequential change detection, and the recognition of dynamic patterns, with some empirical examples.

**Prediction via Preimage:** As is known in physics (nonlinear science), long-term predictions in a nonlinear dynamical system are, in principle, impossible if at least one of its Lyapunov exponents is positive, which would be typically the case of interests. This is true even if the dimension of the system is low because uncertainty involved in the evolution of the system exponentially increases over time. However, it may be possible to predict an observable in the near future (i.e., short-term prediction) if we could formulate a precise predictive model. Therefore, we here consider a prediction based on estimated Koopman spectra as in Eq. (13). Since Eq. (13) is represented as the linear combination of $\phi(\boldsymbol{x}_i)$ ($i = 0, \ldots, \tau - 1$), a prediction can be obtained by considering the pre-image of the predicted observables in the feature space. Even though any method for finding a pre-image of a vector in the feature space can be used for this purpose, here we describe an approach

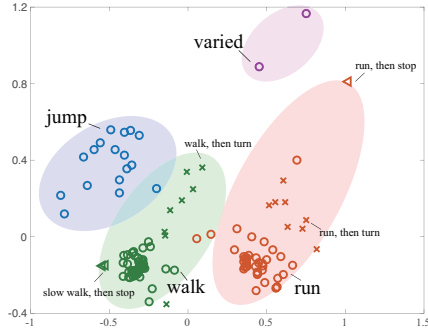

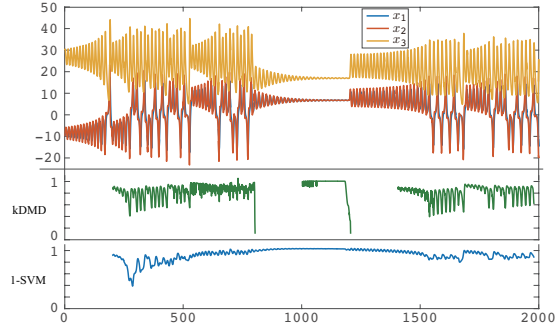

Figure 3: MDS embedding with the distance matrix from kernel principal angle between subspaces of the estimated Koopman eigenfunctions for locomotion data. Each point is colored according to its assigned motion (jump, walk, run, and varied).

Figure 4: Sample sequence (top) and change scores by our method (green) and the kernel change detection method (blue).

based on a similar idea with multidimensional scaling (MDS), as describe in [18], where a pre-image is recovered to preserve the distance between it and other data points in the input space as well as the feature space. The basic steps are (i) find $n$-neighbors of a new point $\hat{\phi}(\boldsymbol{x}_{\tau+l})$ in the feature space, (ii) calculate the corresponding distance between the preimage $\hat{\boldsymbol{x}}_{\tau+l}$ and each data point $\boldsymbol{x}_t$ based on the relation between the feature- and input-space distances, and (iii) calculate the pre-image in order to preserve the input distances. For step (i), we need the distance between the estimated feature and each data point in the feature space, which is calculated as

$$\|\hat{\phi}(\boldsymbol{x}_{\tau+l}) - \phi(\boldsymbol{x}_t)\|^2 = \|\hat{\phi}(\boldsymbol{x}_{\tau+l})\|^2 + \|\phi(\boldsymbol{x}_t)\|^2 - 2\hat{\phi}(\boldsymbol{x}_{\tau+l})^*\phi(\boldsymbol{x}_t)$$
$$= \boldsymbol{c}^*(\mathcal{M}_\tau^*\mathcal{M}_\tau)\boldsymbol{c} + k(\boldsymbol{x}_t, \boldsymbol{x}_t) - 2\boldsymbol{c}^*(\mathcal{M}_\tau^*\phi(\boldsymbol{x}_t)),$$

where $\boldsymbol{c}$ is from Eq. (10). Note that the first and third terms in the above equation can be calculated using the values in the Gram matrix for the data. Once we obtain $n$-neighbors based on the feature distances, we can construct the corresponding local coordinate by calculating a set of orthogonal bases (via, for example, singular value decomposition of the data matrix for the neighbors) based on the distances in the input spaces, which are analytically obtained from the feature distances [18]. The graphs in Fig. 2 show empirical examples of the true versus predicted values as described above for the toy nonlinear system and the Hénon map. The setups for the data generation and the kernels etc. are same with the previous section.

**Embedding and Recognition of Dynamics:** A direct but important application of the presented analysis is the embedding and recognition of dynamics with the extracted features. Like (kernel) PCA, a set of Koopman eigenfunctions estimated via the analysis can be used as the bases of a low dimensional subspace that represents the dynamics. For example, the recognition of dynamics based on this representation can be performed as follows. Suppose we are given $m$ collection of data sequences $\{\boldsymbol{x}_t\}_{t=0}^{T_i}$ ($i=1,\dots,m$) each of which is generated from some known dynamics $C$ (e.g., walks, runs, jumps etc.). Then, a set of estimated Koopman eigenfunctions for each known dynamics, which we denote by $\mathcal{A}_c = \mathcal{M}_\tau \boldsymbol{w}_c$ for the corresponding complex vector $\boldsymbol{w}_c$, can be regarded as the bases of a low-dimensional embedding of the sequences. Hence, if we let $\mathcal{A}$ be a set of the estimated Koopman eigenfunctions for a new sequence, its category of dynamics can be estimated as

$$\hat{i} = \underset{c \in C}{\operatorname{argmin}} \operatorname{dist}(\mathcal{A}, \mathcal{A}_c),$$

where $\operatorname{dist}(\mathcal{A}, \mathcal{A}_c)$ is a distance between two subspaces spanned by $\mathcal{A}$ and $\mathcal{A}_c$. For example, such a distance can be given via the kernel principal angles between two subspaces in the feature space [36]. Fig. 3 shows an empirical example of this application using the locomotion data from CMU Graphics Lab Motion Capture Database.[2] We used the RBF Gaussian kernel, where the kernel width was set as the median of the distances from a data matrix. The figure shows an embedding of the sequences via MDS with the distance matrix, which was calculated with kernel principal angles [36] between subspaces spanned by the Koopman eigenfunctions. Each point is colored according to its motion (jump, walk, run, and varied).

**Sequential Change-Point Detection:** Another possible application is the sequential detection of change-points in a nonlinear dynamical system based on the prediction via the presented analysis. Here, we give a criterion for this problem based on the so-called cumulative-sum (CUSUM) of likelihood-ratios (see, for example, [9]). Let $\boldsymbol{x}_0, \boldsymbol{x}_1, \boldsymbol{x}_2, \ldots$ be a sequence of random vectors distributed according to some distribution $p_h$ ($h = 0, 1$). Then, change-point detection is defined as the sequential decision between hypotheses; $\mathcal{H}_0\colon p(\boldsymbol{x}_i) = p_0(\boldsymbol{x}_i)$ for $i = 1, \ldots, T$, and $\mathcal{H}_1\colon p(\boldsymbol{x}_i) = p_0(\boldsymbol{x}_i)$ for $i = 1, \ldots, \tau$ and $p(\boldsymbol{x}_i) = p_1(\boldsymbol{x}_i)$ for $i = \tau + 1, \ldots, T$, where $1 \leq \tau \leq T(\leq \infty)$. In CUSUM, the stopping rule is given as

$$T^* = \inf \left\{ T : \max_{1 \leq \tau < T} \sum_{t=\tau+1}^{T} \log \left( p_1(\boldsymbol{x}_t)/p_0(\boldsymbol{x}_t) \right) \geq c \right\},$$

where $c > 0$ ($T^*$ is the stopping time). Although the Koopman operator is, in general, defined for a deterministic system, it is known to be extended to a stochastic system $\boldsymbol{x}_{t+1} = f(\boldsymbol{x}_t, \boldsymbol{v}_t)$, where $\boldsymbol{v}_t$ is a stochastic disturbance [20]. In that case, the operator works on the expectation. Hence, let us define the distribution of $\boldsymbol{x}_t$ as a nonparametric exponential family [7], given by

$$p(\boldsymbol{x}_t) = \exp \left( \langle \theta(\cdot), \psi(\boldsymbol{x}_t) \rangle_{\mathcal{H}} - g(\theta) \right) = \exp \left( \langle \phi \circ \boldsymbol{f}(\boldsymbol{x}_{t-1}), \phi(\boldsymbol{x}_t) \rangle_{\mathcal{H}} - g(\phi \circ \boldsymbol{f}(\boldsymbol{x}_{t-1})) \right),$$

where $g$ is the log-partition function. Then, the log-likelihood ratio score is given as

$$\log \Lambda_\tau(\boldsymbol{x}_{1:T}) := \sum_{i=\tau+1}^{T} \log \left( p_1(\boldsymbol{x}_t)/p_0(\boldsymbol{x}_t) \right) \propto -\sum_{i=\tau+1}^{T} \left( \sum_{j=1}^{\tau} \alpha_i^{(0)} k(\boldsymbol{x}_j, \boldsymbol{x}_i) - \sum_{j=1}^{\tau} \alpha_i^{(1)} k(\boldsymbol{x}_j, \boldsymbol{x}_i) \right),$$

where $\alpha_i^{(0)}$ and $\alpha_i^{(1)}$ are the coefficients obtained by the proposed algorithm with the data for $i = 1, \ldots, \tau$ and $i = \tau + 1, \ldots, T$, respectively. Here, since the variation of the second term is much smaller than the first one (cf. [7]), the decision rule, $\log \Lambda_* \geq c$, can be simplified by ignoring the second term. As a result, we have the following decision rule with some critical value $\tilde{c} \leq 0$:

$$-\log \Lambda_\tau(\boldsymbol{x}_{1:T}) \approx \sum_{i=\tau+1}^{T} \sum_{j=1}^{\tau} \alpha_i^{(0)} k(\boldsymbol{x}_j, \boldsymbol{x}_i) \leq \tilde{c},$$

A change-point is detected if the above rule is satisfied. Otherwise, the procedure will be repeated until a change-point is detected by updating the coefficients using new samples. Fig. 4 shows an empirical example of the (normalized) change score calculated with the proposed algorithm, with comparison with the one by the kernel change detection method (cf. [7]), for the shown data generated from the Lorenz map. We used the RBF Gaussian kernel as in the same way. In the simulation, the parameter of the map changes at 800 and 1200 although the ranges of the data values dramatically change in other areas (where the score by the comparative method has changed correspondingly).

# 8   Conclusions

We presented a spectral analysis method with the Koopman operator in RKHSs, and developed algorithms to perform the analysis using a finite-length data sequence from a nonlinear dynamical system, that is essentially reduced to the calculation of a set of orthogonal bases of the Krylov matrix in RKHSs and the eigendecomposition of the projection of the Koopman operator onto the subspace spanned by the bases. We further considered applications using estimated Koopman spectra with the proposed analysis, which were empirically illustrated using synthetic and real-world data.

**Acknowledgments**

This work was supported by JSPS KAKENHI Grant Number JP16H01548.

## Footnotes

[1] The Matlab code is available at `http://en.44nobu.net/codes/kdmd.zip`

[2]Available at `http://mocap.cs.cmu.edu`.

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
