[Reviews · NeurIPS 2016]

Reviewer 1

Summary

This paper introduced a spectral analysis using the Koopman operator defined in a proper reproducing kernel Hilber space. Then they followed the standard pathway to define a set of (kernel) approaches to do analysis of nonlinear dynamical systems with empirical data. The eigendecomposition of the projection of the Koopman operator resulted in some efficiency gain in both synthetic and real data analysis problems.

Qualitative Assessment

This paper introduces a significant kernel construction to analyze nonlinear dynamical systems. The contribution is well motivated and experimentally validated. There are, however, some concerns that I hope the authors may address: * It is not clear to me the (hopefully substantial) diference between what is presented here, and the paper [35] M.O. Williams, I.G. Kevrekidis, and C.W. Rowley. A data-driven approximation of the Koopman operator: Extending dynamic mode decomposition. Journal of Nonlinear Science, 25:1307–1346, 2015, where the Koopman eigenvalues, eigenfunctions, and modes were already introduced to analyze nonlinear system dynamics. I'd appreciate some comments and discussion pointing out and clarifying the main differences, as well as an improved discussion on section 4. * The field of nonlinear dynamical system analysis with kernels is vast, and some relevant references are missing; authors could check papers by L. Ralaivola and F. d’Alche Buc, Tuia et al, Rojo-Alvarez et al., M. Mouattamid and R. Schaback, or M. O. Franz and B. B. Schölkopf. * Even though the Koopman operator is, in general, defined for a deterministic system, I'd appreciate some comments on the performance on stochastic settings, which are after all the important/realistic ones. Authors briefly touch upon this issue, but some further discussion, and eventually a toy experiment would be welcome. * Authors clarify the remark (line 216, page 5) that "the estimation procedure itself does not necessarily require a sequence but rather a collection of pairs of consecutive observables, where each pair is supposed to be" nonlinear pointwise related. I'm curious about the implication of this assumption in multidimensional time series with eventually weak pointwise relations: what do we gain with the proposed kernel construction? Could you discuss on this? Related to this, I'd also appreciate if authors can elaborate on the causal assumption inherent in the setting. * With regard the experimental settings, I sincerely find them not very ambitious: essentially prediction via preimaing follows a standard well-known approach, under the "Embedding and Recognition of Dynamics" subsection there is a lack of discussion on the connections (and not to mention experimental comparison) between the proposed approach and the (vast) literature of clustering time series (e.g. through kernel k-means and related subspace kernel methods), and when it comes to the sequential change-point detection experiment it would be more appealing to see a comparison to state of the art online change detection with kernels (e.g. the one used by the authors is known to fail in complex dynamics). Some more convincing experiments are expected to support the development. ** Minor: the paper is very well written and clear. I just spotted two minor typos: please check indices on tau-1 vs. tau, and replace dot with comma in k(xi.xj) (last eq in p7)

Confidence in this Review

2-Confident (read it all; understood it all reasonably well)


Reviewer 2

Summary

The paper proposed a spectral method for the analysis of nonlinear dynamical systems. The so-called Koopman operator is an operator representation of function composition. In this case, the Koopman operator decomposition is applied to the forward nonlinear dynamics of the system, x(t+2) = f(f(x(t))). Given collected trajectories from the system, then one can perform some kind of eigen-decomposition of the operator using similar idea from kernel PCA and subspace analysis. Last the method is applied to embedding and recognition of dynamical systems, and the author show that the method performs well.

Qualitative Assessment

To an audience who is familiar with kernel methods, the new things in the paper is that the composition of functions can be represented as a linear operator and some interesting decomposition can be applied to function compositions. Dynamical system is a particular composition of functions, and a good application of this. Since the algorithm, dynamic mode decomposition (DMD), has been proposed already in dynamical system literature, the novelty of the paper seems to be limited to kernelize it. There are other methods for recognizing dynamical system, such as recurrent neural works, and for detection changes. In a sense the applications in the paper is not very compelling to demonstrate why the method is good and unique.

Confidence in this Review

2-Confident (read it all; understood it all reasonably well)


Reviewer 3

Summary

This paper derives a spectral analysis of the Koopman (composition) operator in RKHSs and presents a modal decomposition algorithm to perform the spectral analysis. The authors also discuss a robustifying method and perform several numerical experiments to demonstrate the usefulness of their approach. The overall framework is an extension of "dynamical mode decomposition" using the kernel trick.

Qualitative Assessment

Strength of the paper: The paper is very well written and organized. The authors present clearly the different theoretical concepts used in the derivation of the method, which seems very interesting to me. Moreover, it is technically solid. There is no deep comparison with state-of-the-art approaches but numerical experiments illustrate correctly the usefulness of the method. It is sufficient in my view. However, there are some disturbing points explained below. Major flaws: As for me, the major flaw of this paper is Section 3.1 and particularly Equation (7) as well as the proof of Theorem 1, that I do not fully understand. On one hand, Equation (7) is similar to Equation (2), so what is the purpose of it ? On the other hand, Equation (7) does not seem consistent to me since K_H is a linear operator from H to itself and \phi is definitely not in H (this is a function from M to H). Can the authors explain this point in more details ? This gives me some doubts concerning the proof Lines 144-157. In addition, what does "by applying the operator K_H to both sides" mean (Line 149) ? Since both sides are scalar numbers, then K_H : H \to H does not apply. The assumptions in Theorem 1 look strong (and not very explicit). Could the authors explain these assumptions ? Moreover, additional assumptions appear in the proof of the theorem. Why ? Can they hold easily ? Typos: - Line 55: the the - Line 153: \kappa_j / \lambda_j instead of \kappa_i / \lambda_i

Confidence in this Review

2-Confident (read it all; understood it all reasonably well)


Reviewer 4

Summary

This paper extends Koopman spectral analysis to reproducing kernels. Method is evaulated on simulated and real data.

Qualitative Assessment

This is a very well written paper. I would have liked to see two additional aspects to this paper, the first being a comparison to non-RKHS Koopman spectral analysis in the simulations. The second (and this could be explored in simulations as well) would be a discussion/exploration of choice of kernels, and how that could affect results.

Confidence in this Review

2-Confident (read it all; understood it all reasonably well)


Reviewer 5

Summary

The manuscript, entitled “Koopman Spectral Analysis of Nonlinear Dynamical Sys- tems with Reproducing Kernels” extended the dynamical mode decomposition (DMD) to reproducing kernel Hilbert space (RKHS). The resulting algorithm (KDMD) is then applied to a few toy problems. It hence allows to use features map as definition of the kernel, in order to link more the DMD to specific features of interest; it is then a powerful generalization of DMD. My main concern is that the paper is really hard to read. Even for a few simple notions, the authors emphasize on the mathematical writings when short sentences would have been more than enough. Some sections, on the other hand, are actually verbose. It might make it difficult to understand even for specialists, and a few points should be clarified. It is too cryptic for publication in the Neural Information Processing Systems conference proceedings, but I strongly encourage the authors to rewrite it in a more pedagogic way; the resulting article should actually be highly cited.

Qualitative Assessment

Referee’s report on: Koopman Spectral Analysis of Nonlinear Dynamical Systems with Reproducing Kernels The manuscript, entitled “Koopman Spectral Analysis of Nonlinear Dynamical Sys- tems with Reproducing Kernels” extended the dynamical mode decomposition (DMD) to reproducing kernel Hilbert space (RKHS). The resulting algorithm (KDMD) is then applied to a few toy problems. It hence allows to use features map as definition of the kernel, in order to link more the DMD to specific features of interest; it is then a powerful generalization of DMD. My main concern is that the paper is really hard to read. Even for a few simple notions, they emphasize on the mathematical writings when short sentences would have been more than enough. Some sections, on the other hand, are actually verbose. It might make it difficult to understand even for specialists, and a few points should be clarified. It is too cryptic for publication in the Neural Information Processing Systems conference proceedings, but I strongly encourage the authors to rewrite it in a more pedagogic way; the resulting article should actually be highly cited. I. MAJOR REMARKS Please find my major remarks in the following: 1. The presentation is really verbose when it could be simplified. For instance: “Con- ceptually, DMD can be considered as producing an approximation of the Koopman eigenfunctions using a set of linear monomials of observables as basis functions, which is analogous to a one-term Taylor expansion at each point.” can be easily summarized in one short and simple sentence. 2. Sec. 4 is actually useless in a short paper and was (partially) treated in the introduc- tion. 3. Sec. 5 should have been illustrated, or be summarized in the conclusion. 4. Sec. 6 should have been developed in the way of Sec. 5! It has been condensed too much. II. MINOR REMARKS My other minor remarks are: 1. there is probably a typo in the description of the toy NL case, as it is actually linear. 2. first two problems are actually polynomial. It is were the KDMD should shine - when the kernel can be guessed, as in 1 . More details should be given, as actually in Sec. 5. REFERENCES 1 Steven L. Brunton, Bingni W. Brunton, Joshua L. Proctor, J. Nathan Kutz, “Koopman invariant subspaces and finite linear representations of nonlinear dynamical systems for control,” http://arxiv.org/abs/1510.03007v2, (2015)

Confidence in this Review

3-Expert (read the paper in detail, know the area, quite certain of my opinion)


Reviewer 6

Summary

In the present paper the authors derive a spectral analysis of the Koopman operator in reproducing kernel Hilbert spaces and develop an algorithm to perform this analysis based on measurement data from a dynamical system. The Koopman operator is a concept from fluid dynamics, and the authors seem to imply it has relevance to machine learning. The authors demonstrate the algorithm and the analysis using two simple dynamical systems a damped linear 1D dynamics and a chaotic map. They also apply their method to real-world locomotion data and compare the performance with standard dynamic mode decomposition, PCA and kernel PCA, obtaining somewhat unclear results.

Qualitative Assessment

The paper derives an interesting extension of dynamic mode decomposition, somewhat along the lines of older work on "kernelizing" techniques phrased in terms of linear operators (such as kernel PCA, kernel ICA, kernel independence tests, GP bandits, etc.). The authors provide some experimental results on largely artificial problems. The paper is overall clearly written, and has educational value for people foreign to concepts used in fluid dynamics. I am in general supportive of work which introduces machine learners to techniques in different disciplines. Here, this seems a decomposition technique for functions from M to M, that are used to construct a dynamical system (without any stochastic components). I am personally not aware of such setups being used in machine learning (typically, people use dynamical systems where random effects play a role, like innovations and noise), but I may miss something. In the current case, I think the authors need to do more work to convince ML readers of the practical usefulness of these techniques. What are potential use cases? Can you compete, at least to some extent, with the state of the art there? An ML discipline where inference and learning in dynamical systems is important, and where a lot of work has been done, is robotics. What would Koopman analysis buy somebody working in robotics? Or please pick any other domain. And while the theoretical derivations are well executed, I am not terribly surprised about any of them, given I know about kernel PCA or kernel ICA, or follow-up work on kernel independence tests or even "kernel Bayes rule". You have a linear operator, and you can do nice things if functions live in an RKHS. You end up with algorithms that do some sort of eigendecomposition on finite-D matrices. Some details: It is not quite clear to me, whether in the modal decomposition algorithm one needs the explicit representation in feature space. If so, this is a limitation compared to kernel PCA and the authors should make this more explicit. It would also be good if the authors would show the performance of standard dynamic mode decomposition on the toy example and the Henon map to demonstrate the differences in this simple setup.

Confidence in this Review

2-Confident (read it all; understood it all reasonably well)